# Synthesis of Pyridoxine-Derived Dimethylpyridinols Fused with Aminooxazole, Aminoimidazole, and Aminopyrrole

**DOI:** 10.3390/molecules27072075

**Published:** 2022-03-23

**Authors:** Bhuwan Prasad Awasthi, Hyunji Lee, Byeong-Seon Jeong

**Affiliations:** 1College of Pharmacy, Yeungnam University, Gyeongsan 38541, Korea; awasthi.bhuwan2047@gmail.com; 2Department of Chemistry and Howard Hughes Medical Institute, University of Illinois at Urbana-Champaign, 600 South Mathews Avenue, Urbana, IL 61801, USA

**Keywords:** aminobicyclic pyridinols, pyridoxine, oxazole, imidazole, pyrrole

## Abstract

Described in this paper are studies on the preparation of three classes of dimethylpyridinols derived from pyridoxine fused with aminooxazole, aminoimidazole, and aminopyrrole. The key feature of this synthetic strategy is the manipulation of hydroxymethyl moiety of C(5)-position of the pyridoxine starting material along with the installation of an amino group at C(6)-position. Efficient and practical synthesis for the oxazole- and imidazole-fused targets was accomplished, while the instability of the pyrrole-fused one was observed.

## 1. Introduction

Phenotypic screening campaigns have long been crucial to the identification of biologically active small molecules in conjunction with target-based approaches [1,2]. Because of their biological relevancy and favorable physicochemical properties, novel scaffolds from nature are frequently used to inspire the preparation of compound collections for phenotypic screening [3,4,5,6]. Pyridoxine (**1**) is a form of vitamin B_6_ that scavenges reactive oxygen species and regulates diverse cellular metabolisms, including amino acid biosynthesis and fatty acid biosynthesis (Figure 1) [7,8,9,10,11]. Inspired by pyridoxin (**1**), our group has pursued derivative syntheses along with biological investigations to explore the pharmacological potential of the 6-aminopyridin-3-ol scaffold (**2**). The class of aminopyridinols (**3**) was found to have anti-angiogenic activity, which is linked to cancer and age-related diseases, as well as anti-inflammatory activity against ulcerative colitis. [12,13]. Notably, remarkable anticancer activity was observed in novel structures where the 6-aminopyridin-3-ol core was hybridized with biologically active agents such as α-tocopherol and sunitinib. An α-tocopherol-hybridized compound **10** suppressed lung tumor growth by dual inhibition of NADPH oxidase 2 and receptor tyrosine kinases. A sunitinib-hybridized compound **11** was also discovered as an apoptosis-inducing anticancer agent by controlling the p53 level, with a safer cytotoxicity profile than sunitinib [14]. Therefore, these results showcase that the 6-amino-pyridin-3-ol scaffolds indeed have the potential to provide favorable biological activities and demand further studies to expand the scope of the scaffolds.

Our previous synthetic strategies for lead compound discovery are summarized and illustrated in Figure 1 [12,13,15,16,17,18,19,20,21,22]. Using an amine group of **2** at C(6)-position as a synthetic handle, amidopyridinols **3** and ureido-/thioureido-/carbamato-pyridinols **4** have been synthesized to generate a diverse set of compound collections [23,24,25,26,27]. In addition, heteroatom-containing bicyclic pyridinols **6** were designed and constructed, providing more constrained structures [14,28,29,30,31,32,33]. As an ongoing project to increase the compound diversity of 6-amino-pyridin-3-ol collections, here we present synthetic studies towards the preparation of three bicyclic pyridinol backbones **12**~**14** fused with five-membered heteroaromatic ring systems (Figure 1). Our strategy for the synthesis of pyrrolo- (**12**), imidazolo- (**13**), and oxazolo-pyridinol (**14**) features divergent synthesis starting from pyridoxine, in line with the previous syntheses of pyridinol derivatives [31]. We also envision the introduction of an additional amino functional group on a five-membered ring, which would allow for appendage derivatization in the future. In this work, the hydroxy group at C(5′)-position in pyridoxine (**1**) serves as a synthetic tool for the installation of various functionalities, including nitrile, amide, phenolic OH, and anilinic NH_2_ in synthetic intermediates.

## 2. Results and Discussion

A general outline of the synthetic strategy for the preparation of dimethylpyridinols fused with aminooxazole (**12**), aminoimidazole (**13**), and aminopyrrole (**14**) is shown in Figure 2. This began with the preparation of known primary alcohol **15**, which was used as a common intermediate for the syntheses of three final products, according to the three-step sequence starting from pyridoxine (**1**) established by us [10,31].

First, for the synthesis of the aminooxazole-containing analogue **12**, the hydroxymethyl moiety of **15** was converted to amine **17** in three steps [31]. The amino group of **17** was replaced by hydroxy group to afford **18** as we previously reported [31]. Then, an amine group was successfully introduced at the C(2)-position of **18** in the presence of free alcohol at the C(3)-position, without additional protection/deprotection steps [31]. The resulting 2-amino-3-hydroxy pyridinol **19** was nicely constructed to test the formation of the aminooxazole moiety. Gratefully, the formation of the aminooxazole ring was accomplished by using cyanogen bromide in almost quantitative yield (compound **23**) [34], which was followed by debenzylation to afford the first target compound **12** (Figure 3).

Having observed the successful aminooxazole formation, we applied a similar approach to achieve analogue synthesis, which is the installation of corresponding functional groups followed by cyclization as outlined in Figure 2. Since aminopyridinol intermediates (**15**~**17**) with diverse functionalities were generated during the preparation of aminooxazolopyridinol **12**, we envisaged that these intermediates could serve as great branching points for the synthesis of aminoimidazole **13** and aminopyrrole **14**.

The synthesis of aminoimidazolopyridinol **13** initiated with the preparation of the diamine compound **20** (Figure 4). We first attempted to introduce an amino group at 2-position of the pyridine ring in the primary amide compound **16** via phthalimidation and subsequent Hofmann rearrangement (the left side of Figure 4). After *m*-CPBA oxidation of **16** to pyridine *N*-oxide **24**, the nucleophilic addition of phthalimide to the *O*-*p*-toluenesulfonylated pyridinium intermediate followed by the elimination of *p*-toluenesulfonic acid afforded **25**. Removal of the *N*-phthalyl group in **25** was smoothly proceeded by the treatment of hydrazine to give free amine **26** [31]. The trial for Hofmann rearrangement of the primary amide compound **26** to obtain the diamine compound **20** was performed in a mixed solvent with alkaline water and tetrahydrofuran, employing hypochlorite for activation of the primary amide. However, contrary to our expectations, the desired diamine compound **20** was not produced under the reaction conditions. A cyclic urea compound **27** was instead obtained in a moderate yield (40%).

A possible route of the formation of imidazolidinone compound **27** is shown in Figure 5. Once reactive isocyanate intermediate **32** was formed via the rearrangement of *N*-chloroamide intermediate **31**, and the amine at 2-position might attack the isocyanate to form a five-membered cyclic urea **27**. Since we experienced such a fact in this trial, we expected that using water as a solvent would increase the probability of transformation of isocyanate intermediate **32** to carbamic acid **33**, which then spontaneously decomposed to the desired diamine **20** with the liberation of carbon dioxide. However, based on the results we observed, the intramolecular cyclization outcompeted the hydroxide (or water) addition to the isocyanate **32**. Several attempts to cleave the urea of compound **27** to obtain the diamine **20** under acidic conditions with high temperatures only gave a debenzylated compound quantitatively instead of the desired product.

To avoid such a troublesome intramolecular cyclization issue, we decided to change the synthetic sequence. The conversion of the primary amide group at C(3)-position to a free amino group was performed prior to the installation of another free amino group at C(2)-position as depicted on the right side of Figure 4. Hofmann rearrangement of **16** followed by *N*-oxide formation (**28**) and phthalimide substitution proceeded smoothly to afford **29**, which was then treated with hydrazine to produce the diamine compound **20**. The aminoimidazole backbone of compound **30** was successfully constructed by the treatment of **20** with cyanogen bromide. Finally, the aminoimidazole-fused pyridinol **13** was obtained by debenzylation.

Next, we investigated the synthesis of aminopyrrole-containing analog **14** (Figure 6). The initial attempt includes the substitution of primary alcohol of the common intermediate **15** with nitrile functionality (**21**) [31] and phthalimidation of *N*-oxide compound **34**. Reactions proceeded smoothly, generating compound **35**, and phthalimide cleavage using hydrazine successfully furnished the requisite substrate **36** for aminopyrrole formation.

Having installed the 2-amino group together with the nitrile group in **36**, trials were made for the cyclization to form a 2-amino-7-azaindole framework. According to a database search result using the SciFinder^n^ on the transformation of reactants **38** to products **39**, no example has been known in the case of an unsubstituted compound at C(#)-position (Figure 7). Even if the search range was expanded to cases where there was a substituent other than hydrogen at the C(#)-position, only a couple of examples were found including the synthesis of compound **41** [35,36]. Formation of 2-aminoindoles, such as compound **43**, from C(#)-unsubstituted-2-(2-aminopyridin-3-yl)acetonitrile compounds, such as **42**, was also found to be rare [37,38].

The first trial was done under basic conditions using sodium methoxide, which has been successfully applied in the preparation of 2-aminoindole ring formation (Figure 8) [37]. Contrary to the literature in which the 2-aminoindole **43** was obtained in 70% yield from **42** (Figure 7), the reaction of compound **36** under the same reaction conditions was messy, and the desired product **37** was not obtained. It can be presumed that the difference in nucleophilicity between the amine attached to benzene in compound **42** and the amine attached to pyridine in compound **36** might be one of the main reasons for the success or failure of the 2-aminopyrrole cyclization reaction under similar reaction conditions. Interestingly, the formation of a very small amount of a by-product **44** was observed instead. We speculate that this side reaction may have proceeded as follows (Figure 9). Formation of α-ketonitrile intermediate **47** might have been facilitated by the deprotonation with methoxide at the site adjacent to the nitrile group in **36**, which is also a benzylic position. After the successive aerobic oxidation, the cyano group of **47** might be replaced by methoxide anion affording compound **44**.

To avoid such a deprotonation/oxidation problem, we exposed compound **36** to hot acetic acid. However, the desired 2-amino-7-azaindole compound **37** was not obtained under the acidic conditions either. Instead, its *N*-acetyl compound **44** was obtained, albeit in small quantities, and the formation of a lactam **46** was also observed. Use of an additive like zinc acetate to activate the cyclization improved the formation of compound **44** to some extent. Unlike the other two targets, aminooxazolopyridinol and aminoimidazolopyridinol, which were readily prepared, an aminopyrrolopyridine-containing compound like **37** might be presumed to have quite poor chemical stability. The free amino compound **37** was not obtained under both basic and acidic reaction conditions, and even the isolated pure *N*-acetylated compound **44** was observed to gradually decompose in solution over time. Naturally, all efforts made for *N*-deacetylation were not fruitful.

Based on our observations, a possible mechanism for the generation of **45** and **46** was proposed in Figure 10. The reaction is initiated by intramolecular cyclization of **36** to form a 1,3-dihydro-2*H*-pyrrole-2-imine **37′**, which undergoes tautomerization to yield a 2-aminopyrrole **37**. Under the reflux condition, the electron-rich amino group in **37** further reacts with acetic acid to generate *N*-acetyl adduct **45**. Another pathway involves the nucleophilic addition of acetic acid to the electrophilic imine-carbon in **37′** and the subsequent elimination of ammonia to afford **48**. Acidolysis of the resulting acetate **48** followed by tautomerization provides a lactam **46**. Since high temperature and long reaction time are necessary to make the intramolecular cyclization of **36** happen, the reversible reactions illustrated in Figure 10 might occur and generate the mixture of **45** and **46**. Unlike our system, compound **40** contains an additional ester group at the C(#)-position (Figure 7) [35]. This carbonyl functional group may accelerate cyclization/tautomerization events and make the resulting amine of **41** less nucleophilic, preventing the formation of an *N*-acetyl adduct of **41**.

## 3. Materials and Methods

### 3.1. General

Unless otherwise specified, all the materials obtained from Merck (Kenilworth, NJ, USA), TCI (Tokyo, Japan), and Alfa Aesar (Ward Hill, MA, USA) were used without further purification. Inert gas conditions were applied to the sensitive reactions towards air and moisture. The progress of the reactions was traced by TLC analysis using silica gel 60 F_254_ plates (Merck, Kenilworth, NJ, USA). Visualization of the TLC spots was done using a UV lamp (254 nm, Spectroline Corp., Westbury, NY, USA) and staining solutions (Anisaldehyde solution and KMnO_4_ solution) prepared by us using commercially available reagents. Products were purified by flash column chromatography using silica gel 60 (70–230 mesh, Merck, Kenilworth, NJ, USA) or by using the Biotage ‘Isolera One’ (Biotage, Uppsala, Sweden) system with indicated solvents. Melting points were determined using a Büchi melting point B-540 apparatus (Büchi Labortechnik, Flawil, Switzerland) and were unchanged. ^1^H and ^13^C NMR spectra were obtained using a Bruker-250 spectrometer (^1^H and ^13^C frequencies were 250 and 63 MHz, respectively, Bruker Corp., Billerica, MA, USA), and a Bruker Avance Neo 400 spectrometer (^1^H and ^13^C frequencies were 400 and 100 MHz, respectively, Bruker Corp., Billerica, MA, USA). ^1^H and ^13^C NMR spectra of all synthesized molecules are available in the Appendix A. Chemical shifts (δ) were expressed in ppm calibrated to residual solvent signals and the coupling constant (J) in hertz. HR-ESIMS was performed using a Thermo Scientific Q Exactive hybrid Quadrupole-Orbitrap mass spectrometer (Waltham, MA, USA) coupled to a Thermo Scientific Vanquish UHPLC system (Waltham, MA, USA) at the Core Research Support Center for Natural Products and Medical Materials (CRCNM).

### 3.2. 6-(Benzyloxy)-5,7-dimethyloxazolo[4,5-b]pyridin-2-amine (23)

To a solution of cyanogen bromide (30 mg, 0.12 mmol) in H_2_O (2 mL), we added Compound **19** (13 mg, 0.13 mmol). The resulting mixture was refluxed for 15 min. On completion of reaction, the reaction mixture was cooled to room temperature and neutralized using NaHCO_3._ The solids precipitated out were filtered and washed using CHCl_3_. The organic phase was dried over MgSO_4_, filtered, and concentrated to give **23** (33 mg, 99%) as a white solid. R*_f_* 0.36 (CHCl_3_:MeOH = 15:1); m.p. 153 °C; ^1^H NMR (DMSO-*d*_6_) δ 7.69 (s, 2H), 7.51–7.32 (m, 6H), 4.81 (s, 2H), 2.37 (s, 3H), 2.27 (s, 3H); ^13^C NMR (DMSO-*d*_6_) δ 164.6, 152.7, 145.6, 145.0, 138.2, 137.1, 128.4 (2C), 128.2 (2C), 128.1, 120.5, 74.7, 19.2, 9.5; HRMS (ESI) *m*/*z* [M+H]^+^ calculated for C_15_H_16_N_3_O_2_ 270.1237, found 270.1235.

### 3.3. 2-Amino-5,7-dimethyloxazolo[4,5-b]pyridin-6-ol (12)

To a solution of compound **23** (28 mg, 0.10 mmol) in MeOH (3 mL), we added 10% palladium on activated carbon (5 mg). The mixture was stirred under hydrogen atmosphere at room temperature for 6 h. The mixture was filtered through celite pad and the filtrated was concentrated to give **12** (18 mg, 99%) as a white solid. R*_f_* 0.32 (CHCl_3_:MeOH = 7:1); m.p. 282 °C; ^1^H NMR (DMSO-*d*_6_) δ 8.25 (br s, 1H), 7.43 (s, 2H), 2.32 (s, 3H), 2.22 (s, 3H); ^13^C NMR (DMSO-*d*_6_) δ 163.4, 149.5, 143.6, 139.8, 138.4, 115.8, 19.6, 9.5; HRMS (ESI) *m*/*z* [M+H]^+^ calculated for C_8_H_10_N_3_O_2_ 180.0768, found 180.0767.

### 3.4. 6-Benzyloxy-5,7-dimethyl-1H-imidazo[4,5-b]pyridin-2(3H)-one (27)

To a solution of NaOH (100 mg, 0.37 mmol) in THF–H_2_O (1:1, 8 mL), we added 4% aqueous NaOCl (2 mL) and **26** (45 mg, 1.10 mmol). The mixture was stirred at room temperature for 1 h and then at 90 °C for 2 h. The mixture was cooled to room temperature and diluted with CH_2_Cl_2_. The organic layer was neutralized with saturated aqueous NH_4_Cl solution, and the aqueous layer was extracted with CH_2_Cl_2_. The combined organic solution was washed with brine, dried over MgSO_4_, and concentrated to give **27** (39 mg, 40%) as a white solid. R*_f_* 0.40 (CHCl_3_:MeOH = 10:1); m.p. 295 °C; ^1^H NMR (DMSO-*d*_6_) δ 11.0 (br s, 1H), 7.50–7.36 (m, 4H), 4.77 (s, 2H), 2.33 (s, 3H), 2.20 (s, 3H); ^13^C NMR (DMSO-*d*_6_) δ 155.3, 146.5, 140.9, 139.8, 137.3, 128.4 (2C), 128.2 (2C), 128.1, 121.9, 120.3, 74.5, 18.8, 10.6; HRMS (ESI) *m*/*z* [M+H]^+^ calculated for C_15_H_16_N_3_O_2_ 270.1237, found 270.1243.

### 3.5. 5-Amino-3-(benzyloxy)-2,4-dimethylpyridine 1-oxide (28)

To a solution of compound **17** (100 mg, 0.44 mmol) in CH_2_Cl_2_ (3 mL), we added *m*-CPBA (84 mg, 0.49 mmol). The mixture was stirred at room temperature for 1 h and was diluted with CH_2_Cl_2_. The organic layer was successively washed with saturated aqueous NaHCO_3_ solution and brine. The resulting organic solution was then dried over MgSO_4_, filtered, and concentrated. The residue was purified by silica gel column chromatography (CH_2_Cl_2_:MeOH = 15:1) to give **28** (66 mg, 62%) as a white solid. R*_f_* 0.25 (CHCl_3_:MeOH = 15:1); m.p. 164 °C; MS *m*/*z* 245 [M+H]^+^; ^1^H NMR (CDCl_3_) δ 7.80 (s, 1H), 7.51–7.35 (m, 5H), 4.75 (s, 2H), 4.06 (br s, 2H), 2.36 (s, 3H), 2.02 (s, 3H); ^13^C NMR (CDCl_3_) δ 153.2, 141.7, 136.1, 134.6, 128.8 (2C), 128.6, 128.2 (2C), 123.2, 117.9, 75.8, 11.3, 9.9; HRMS (ESI) *m*/*z* [M+H]^+^ calculated for C_14_H_17_N_2_O_2_ 245.1285, found 245.1282.

### 3.6. 2-(3-Amino-5-(benzyloxy)-4,6-dimethylpyridin-2-yl)isoindoline-1,3-dione (29)

To a solution of compound **28** (66 mg, 0.27 mmol) in anhydrous CH_2_Cl_2_ (15 mL), we added phthalimide (40 mg, 0.27 mmol), *N*,*N*-diisopropylethylamine (0.15 mL, 0.81 mmol), and *p*-tosyl chloride (77 mg, 0.41 mmol). The mixture was stirred under argon atmosphere at room temperature for 1 h. The mixture was diluted with CH_2_Cl_2_, and the organic phase was washed with H_2_O. The organic solution was dried over MgSO_4_, filtered, and concentrated. The residual solid was filtered and washed with Et_2_O to give **29** (73 mg, 72%) as a white solid. R*_f_* 0.41 (CHCl_3_:MeOH = 20:1); m.p. 156 °C; ^1^H NMR (CDCl_3_) δ 7.93 (dd, *J* = 5.2, 2.9 Hz, 2H), 7.85–7.69 (m, 2H), 7.56–7.33 (m, 5H), 4.84 (s, 2H), 3.60 (br s, 2H), 2.46 (s, 3H), 2.16 (s, 3H); ^13^C NMR (CDCl_3_) δ 167.4 (2C), 153.0, 142.3, 138.2, 136.9, 134.6 (2C), 132.3 (2C), 128.8 (2C), 128.4, 128.0 (2C), 127.5, 126.7, 124.0 (2C), 75.1, 19.0, 10.8; HRMS (ESI) *m*/*z* [M+H]^+^ calculated for C_22_H_20_N_3_O_3_ 374.1499, found 374.1494.

### 3.7. 5-(Benzyloxy)-4,6-dimethylpyridine-2,3-diamine (20)

To a solution of compound **29** (143 mg, 0.38 mmol) in THF–EtOH (1:1, 4 mL) we added hydrazine (1.0 mL). The mixture was stirred at room temperature for 12 h, and the solvent was evaporated. The residue was diluted with Et_2_O, and the insoluble solids were filtered off. The filtrate was diluted with CH_2_Cl_2_ and washed with H_2_O. The organic solution was dried over MgSO_4_, filtered, and concentrated to give **20** (68 mg, 73%) as a brown solid. R*_f_* 0.22 (CHCl_3_:MeOH = 20:1); m.p. 129 °C; ^1^H NMR (CDCl_3_) δ 7.50–7.30 (m, 5H), 4.72 (s, 2H), 4.08 (br s, 2H), 3.26 (br s, 2H), 2.33 (s, 3H), 2.09 (s, 3H). ^13^C NMR (CDCl_3_) δ 145.8, 144.0, 138.7, 137.4, 128.7 (2C), 128.2, 128.1 (2C), 126.3, 125.6, 75.4, 18.5, 10.4; HRMS (ESI) *m*/*z* [M+H]^+^ calculated for C_14_H_18_N_3_O 244.1444, found 244.1442.

### 3.8. 6-(Benzyloxy)-5,7-dimethyl-3H-imidazo[4,5-b]pyridin-2-amine (30)

To a solution of cyanogen bromide (57 mg, 0.53 mmol) in H_2_O (2 mL) we added compound **20** (130 mg, 0.53 mmol) in H_2_O (4 mL). The reaction mixture was refluxed for 12 h. The mixture was cooled to room temperature, and the solids precipitated out were filtered and recovered using CHCl_3_. The organic solution was dried over MgSO_4_ and concentrated to give **30** (129 mg, 85%) as a beige solid. R*_f_* 0.24 (CHCl_3_:MeOH = 9:1); ^1^H NMR (DMSO-*d*_6_) δ 7.53–7.47 (m, 2H), 7.45–7.33 (m, 3H), 6.31 (s, 2H), 4.77 (s, 2H), 2.37 (s, 3H), 2.29 (s, 3H); ^13^C NMR (DMSO-*d*_6_) δ 156.1, 145.9, 140.0, 137.6, 128.4 (2C), 128.1 (2C), 127.9, 74.5, 19.1, 10.6; HRMS (ESI) *m*/*z* [M+H]^+^ calculated for C_15_H_17_N_4_O 269.1397, found 269.1393.

### 3.9. 2-Amino-5,7-dimethyl-3H-imidazo[4,5-b]pyridin-6-ol hydrochloride (13)

To a suspension of compound **30** (20 mg, 0.075 mmol) in CH_2_Cl_2_ (3 mL), we added 1 M BCl_3_ in CH_2_Cl_2_ (0.75 mL) at 0 °C. After the mixture was stirred overnight, a mixed solvent of CHCl_3_:MeOH (9:1) was added. The resulting solution was concentrated to give **13** (14 mg, 87%) as a white solid. R*_f_* 0.20 (CHCl_3_:MeOH = 5:1); ^1^H NMR (CD_3_OD) δ 2.60 (s, 3H), 2.53 (s, 3H); ^13^C NMR (CD_3_OD) δ 155.3, 148.2, 136.4, 134.8, 128.9, 127.4, 15.6, 12.0; HRMS (ESI) *m*/*z* [M+H−Cl]^+^ calculated for C_8_H_11_N_4_O 179.0927, found 179.0927.

### 3.10. 3-(Benzyloxy)-5-(cyanomethyl)-2,4-dimethylpyridine 1-oxide (34)

To a solution of compound **21** (100 mg, 0.40 mmol) in CH_2_Cl_2_ (5 mL), we added *m*-CPBA (75 mg, 0.44 mmol), and the resulting mixture was stirred at room temperature for 1 h. The mixture was diluted with CH_2_Cl_2_ and washed with saturated aqueous NaHCO_3_ solution and brine. The organic layer was dried over MgSO_4_, filtered, and concentrated. The residue was purified by silica gel column chromatography (CH_2_Cl_2_:MeOH = 25:1) to give **34** (98 mg, 92%) as a white solid. R*_f_* 0.28 (CHCl_3_:EtOAc = 20:1); m.p. 144 °C; ^1^H NMR (CDCl_3_) δ 8.21 (s, 1H), 7.45–7.35 (m, 5H), 4.83 (s, 2H), 3.60 (s, 2H), 2.47 (s, 4H), 2.21 (s, 3H); ^13^C NMR (CDCl_3_) δ 153.8, 145.3, 135.5, 135.3, 129.9, 129.0, 128.9 (2C), 128.3 (2C), 125.2, 115.6, 76.2, 19.6, 12.1, 12.1; HRMS (ESI) *m*/*z* [M+H]^+^ calculated for C_16_H_17_N_2_O_2_ 269.1285, found 269.1282.

### 3.11. 2-(5-(Benzyloxy)-2-(1,3-dioxoisoindolin-2-yl)-4,6-dimethylpyridin-3-yl)acetonitrile (35)

To a solution of compound **34** (72 mg, 0.27 mmol) in anhydrous CH_2_Cl_2_ (5 mL), we added phthalimide (42 mg, 0.29 mmol) and *p*-tosyl chloride (82 mg, 0.43 mmol). After the reaction mixture became a clear solution, *N*,*N*-diisopropylethylamine (0.15 mL, 0.85 mmol) was added and the resulting mixture was stirred under argon atmosphere at room temperature for 1 h. The mixture was diluted with CH_2_Cl_2_ and washed with H_2_O. The organic solution was dried over MgSO_4_, filtered, and concentrated. The residue was purified by silica gel column chromatography (Hexanes:EtOAc = 5:1) to give **35** (67 mg, 63%) as a white solid. R*_f_* 0.25 (Hexanes:EtOAc = 1:1); m.p. 171 °C; ^1^H NMR (CDCl_3_) δ 7.96 (dd, *J* = 5.5, 3.0 Hz, 2H), 7.80 (dd, *J* = 5.5, 3.1 Hz, 2H), 7.51–7.37 (m, 5H), 4.88 (s, 2H), 3.61 (s, 2H), 2.55 (s, 3H), 2.41 (s, 3H); ^13^C NMR (CDCl_3_) δ 167.0 (2C), 153.6, 152.9, 142.0, 139.3, 136.2, 134.8 (2C), 132.0 (2C), 128.8 (2C), 128.6, 128.0 (2C), 124.2 (2C), 123.3, 115.8, 75.2, 19.7, 17.5, 13.2; HRMS (ESI) *m*/*z* [M+H]^+^ calculated for C_24_H_20_N_3_O_3_ 398.1499, found 398.1497.

### 3.12. 2-(2-Amino-5-(benzyloxy)-4,6-dimethylpyridin-3-yl)acetonitrile (36)

To a solution of compound **35** (265 mg, 0.67 mmol) in THF–EtOH (1:1, 2 mL), we added 60% hydrazine hydrate (2 mL). The reaction mixture was stirred at room temperature for 12 h and concentrated. The residue was diluted with Et_2_O, and the insoluble solids were filtered off. The filtrate was diluted with CH_2_Cl_2_ and was washed with H_2_O. The organic solution was dried over MgSO_4_, filtered, and concentrated to give **36** (119 mg, 67%) as a white solid. R*_f_* 0.25 (CHCl_3_:MeOH = 15:1); m.p. 189 °C; ^1^H NMR (CDCl_3_) δ 7.45–7.36 (m, 5H), 4.73 (s, 2H), 4.36 (br s, 2H), 3.52 (s, 2H), 2.39 (s, 3H), 2.24 (s, 3H); ^13^C NMR (CDCl_3_) δ 152.1, 150.4, 145.8, 140.9, 136.9, 128.8 (2C), 128.4, 128.1 (2C), 116.6, 107.0, 75.5, 19.3, 16.4, 12.9; HRMS (ESI) *m/z* [M+H]^+^ calculated for C_16_H_18_N_3_O 268.1444, found 268.1440.

### 3.13. Methyl 2-amino-5-(benzyloxy)-4,6-dimethylnicotinate (44)

To a solution of compound **36** (30 mg, 0.11 mmol) in MeOH (3 mL), we added 25% NaOMe in MeOH (0.25 mL, 1.1 mmol). The mixture was stirred at 50 °C for 1 h under argon atmosphere and cooled to room temperature. The mixture was poured into water and extracted with EtOAc. The organic solution was dried over MgSO_4_, filtered, and concentrated. The residue was purified by silica gel column chromatography (Hexanes:EtOAc = 1:1) to give **44** (3 mg, 10%) as a white solid. R*_f_* 0.30 (CHCl_3_:MeOH = 15:1); ^1^H NMR (CDCl_3_) δ 7.49–7.32 (m, 5H), 5.85 (s, 2H), 4.71 (s, 2H), 3.89 (s, 3H), 2.42 (s, 3H), 2.37 (s, 3H); ^13^C NMR (CDCl_3_) δ 168.8, 155.6, 155.5, 144.6, 143.8, 137.1, 128.8, 128.4, 128.1, 106.4, 75.3, 51.8, 20.0, 15.3; HRMS (ESI) *m*/*z* [M+H]^+^ calculated for C_16_H_19_N_2_O_3_ 287.1390, found 287.1355.

### 3.14. N-(5-(benzyloxy)-4,6-dimethyl-1H-pyrrolo[2,3-b]pyridin-2-yl)acetamide (45)

To a solution of compound **36** (13 mg, 0.048 mmol) in glacial acetic acid, we added zinc acetate (45 mg, 0.24 mmol). The mixture was refluxed under argon atmosphere for 12 h. The mixture was concentrated, and the residue was diluted with H_2_O. The aqueous layer was extracted with CH_2_Cl_2_, and the combined organic solution was dried over MgSO_4_, filtered, and concentrated. The residue was purified by silica gel preparative thin layer chromatography to give **45** (7 mg, 45%) as a brown solid. R*_f_* 0.29 (Hexanes:EtOAc:MeOH = 10:10:1); ^1^H NMR (CDCl_3_) δ 10.55 (br s, 1H), 7.83 (br s, 1H), 7.55–7.47 (m, 2H), 7.46–7.33 (m, 3H), 5.75 (s, 1H), 4.83 (s, 2H), 2.58 (s, 3H), 2.41 (s, 3H), 2.23 (s, 3H); ^13^C NMR (CDCl_3_) δ 168.3, 147.1, 145.1, 141.0, 137.6, 134.7, 130.6, 128.7 (2C), 128.3, 128.1 (2C), 118.9, 82.6, 75.5, 24.1, 19.7, 12.8; HRMS (ESI) *m/z* [M+H]^+^ calculated for C_18_H_20_N_3_O_2_ 310.1550, found 310.1545.

## 4. Conclusions

In this study, the aminooxazole-, aminoimidazole-, and aminopyrrole-containing pyridinols were divergently synthesized starting from pyridoxine. The synthetic strategy relied on the conversion of hydroxymethyl functionality in the common intermediate **15** into phenol, aniline, and cyanomethyl structures, respectively. The installation of nitrogen-containing functional group at α-position in pyridine was achieved via either diazotization or phthalimidation. These synthetic routes enable the rapid construction of the three different aminobicyclic pyridinol derivatives in an efficient and divergent way. With the two chemical handles including the 5′-hydroxy group and arylamino group, these backbones may serve as a valuable starting point for the generation of diverse sets of compound libraries through appendage diversification.

## Data Availability

Data are contained within the article.

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
