# Peer review of "Synthesis of Pyridoxine-Derived Dimethylpyridinols Fused with Aminooxazole, Aminoimidazole, and Aminopyrrole"

_molecules, 2022, doi:10.3390/molecules27072075_

Round 1

Reviewer 1 Report

In their manuscript Lee, Jeong and Awasthi describe preparation of 3 new compounds, comprising of pyridinol annelated with another heterocyclic ring, namely: aminopyrrole, aminooxazole, aminoimidazole. The Authors used reaction of pyridine derivatives with BrCN to accomplish the annulation of the second ring. The approach is not entirely new (see e.g. Org. Prep. Proc. Int., 2021, 528; J. Heterocyclic. Chem. 1976, 1103; J. Heterocyclic. Chem. 2003, 569; Chem. Eur. J. 2020, 13002). However, it is the first time, it was applied for 5-[protected]hydroxypyridines. Syntheses of 2-aminooxazolo[4,5-b]pyridine and 2-aminoimidazo[4,5-b]pyridine were successfully accomplished, while for pyrrolo[2,3-b]pyridine only 2-acetylamino-derivative was obtained. Target compound may be useful for subsequent modifications and elucidation of biological activity. In Reviewer’s opinion the manuscript meets minimum requirements to be published. Major revision is required.

Major issues are:

  • Obtained compounds were not characterized as is usually required. Elemental analysis or high resolution mass spectra are not presented. Therefore brutto-formulas cannot be considered as adequately supported.
  • Many Figures show reactions. Therefore, they should be called “Scheme”.
  • Fig.2. “2 steps” and “1 step” repeated many times is confusing. Why not to write reagents and put a literature reference? Transformation 12->14 takes 3 steps (possibly, all are known in literature). What was the reason to show intermediate 13? Why not to show 12-> in 3 steps?
  • Naming target compounds as 6A, 6B and 6C is somewhat confusing. Please, use usual numeration, i.e., 6,7 and 8.
  • Scheme 2. Upper side shows numerous reactions without reagents, conditions and so on. The Authors should either provide conditions (like it’s done for cases a-f) or put a literature reference above the reaction arrow to indicate, that literature conditions were used.
  • Numeration should be thoroughly checked and corrected. For instance, compounds 20 in Schemes 1 and 2 differs.
  • Why debenzylation to 6A was accomplished with H2,Pd/C, while for 6B BCl3 was employed?
  • The Authors could cite Tetrahedron Lett. 2014, 55, 1296, where synthesis of 2‑aminooxazolo[4,5-b]pyridine is described. There Authors used isothiocyanates, to produce N-substituted derivatives. In the presented manuscript BrCN is used to give NH2-derivatives.
  • line 139. The Reviewer is curious why the Authors compared their approach toward 2-amino-pyrrolo[3,2-b]pyridines with the preparation of aminoindoles. There is literature precedent for this particular type of heterocycles, Chem. Eur. J. 2020, 13002.
  • Check the NMR description. For instance, compound 31: “7.94 (td, J = 5.2, 2.0 Hz, 3H), 7.80 (dd, J = 5.5, 3.1 Hz, 3H).” – there are no three identical protons in the structure. There should be 9 aromatic protons, while (line 264) there are 3+3+9=15 protons is assigned. Is it a misprint or is compound impure? Compound 32 “7.41 (tdd, J = 10.6, 7.5, 5.9 Hz, 5H),” – 5 protons of phenyl group cannot be a tdd.

Minor remarks:

  • line 52. C(5’) position is better to be marked in the Scheme.
  • Fig.1, Now compound 2 is placed above the arrow. It is unclear, whether reactions 1->2->3,4,5,6 were really accomplished or compound 2 is just a general imaginary building block. Please, revise.
  • Compounds 5 and 6. A=C,N,O without valency is incorrect. CH2, NH, O would be better.
  • Fig.3. What is c-HCl? Is it conc. aq. HCl?
  • Nomenclature “N-α-amino-” is somewhat confusing. Why not simply call “α-amino-“?
  • line 301. cyanomethyl, not acetonitrile.
  • Please, provide references for the preparation of starting compounds.
  • line 241, indicate solvent for 1M BCl3.

Reviewer 2 Report

In this report, the authors synthesized different dimethylpyridinols having aminooxazole, aminoimidazole, and aminopyrrole from pyridoxine and they expressed this synthetic strategy is the manipulation of hydroxymethyl moiety of C(5)-position of pyridoxine starting material along with the installation of an amino group at C(6)-position. Moreover, theses pyridoxines cores might be pharmaceutically important and biologically worthy. This referee feels, overall, the scholarly quality of this paper is not satisfactory for acceptance in this journal in the present form. However, there are some further remarks that requisite to be clarify before its publishing.

My comments, suggestions, and questions to authors:

  1. The manuscript holds potentially interesting data but it is not well organized and contains a lot of severances.
  2. I strongly recommend authors to move figure 1 as separate and previous and this report should be in results part with proper citations.
  3. Line 49, citations should be include for previous works.
  4. Line 121, should be rephrase and line 152 ---table 1, entry 2
  5. Table 1 must be corrected, since shown one product and authors obtained different product under different conditions.
  6. For compound 34, does author check this one under inert atmosphere??
  7. In SI file, I can see that data not consisted with their NMR spectrums---compound 24, as well as integrations should be thoroughly checked/compound 6A –8.25 ** broad singlet, 1H
  8. For example, compound 25/3.64 (s, 2H) should be integrated and NMR nuclei whether 250 or 400 MHz should be mentioned.
  9. I recommend authors to double check NMR data and minor typos.

Reviewer 3 Report

This work by Awasthi and colleagues describes the synthesis of some fused heterocycles starting from pyridoxine. This synthetic work is worth publishing, however there are some major points that the authors should address in any subsequent revision:

  1. The English style should be revised in depth. I suggest the authors to use the English Editing Services of MDPI: https://www.mdpi.com/authors/english
  2. The compounds must be analyzed by HRMS (four decimal places), comparing with the theoretical data. The mass data they give are not sufficient for their characterization.
  3. IR spectra of the compounds should be performed, including the data in the section 3 of the manuscript.
  4. It would be recommendable that authors crystallize some of the final products.
  5. 13C-NMR should be performed.
  6. The spectroscopic data of compound 23 should be included in the manuscript and the supplementary material.

Round 2

Reviewer 1 Report

Manuscript appears better after the revision. It can be published after minor corrections:

  • Figure 2 should be called “Scheme”.
  • 2. Naming PyridoxineHCl as Vitamin B6 is somewhat incorrect, since Vitamin B6 is not a hydrochloride salt. The Reviewer would suggest the Authors to remove HCl from the Scheme - it does not influence overall reaction presentation.
  • Check citing of the references. In lines 75-79 ref.[31] is cited three times. Oppositely, refs.[32-33] seems to be not cited in the manuscript.
  • 35. Correct abbreviation is Chem. Eur. J., not Chemistry.

Reviewer 2 Report

the suggested remarks has been incorporated by author and I recommend that the revised manuscript is suitable for publication.

Author Response

Thanks for your evaluation.

Reviewer 3 Report

I consider that the manuscript is suitable for publication in Molecules in its current form.

Author Response

Thanks for your evaluation.